# Organocatalytic Enantioselective Michael Reaction of Aminomaleimides with Nitroolefins Catalyzed by Takemoto’s Catalyst

**DOI:** 10.3390/molecules27227787

**Published:** 2022-11-12

**Authors:** Hongwen Mu, Yan Jin, Rongrong Zhao, Liming Wang, Ying Jin

**Affiliations:** 1Department of Pharmacy, Jilin Medical University, Jilin 132013, China; 2College of Science, Yanbian University, Yanji 133000, China

**Keywords:** Takemoto’s catalyst, enantioselective, Michael addition, aminomaleimides, nitroolefins

## Abstract

Known as electrophiles, maleimides are often used as acceptors in Michael additions to produce succinimides. However, reactions with maleimides as nucleophiles for enantioselective functionalization are only rarely performed. In this paper, a series of bifunctional Takemoto’s catalysts were used to organocatalyze the enantioselective Michael reaction of aminomaleimides with nitroolefins. The resulting products were obtained in good yields (76–86%) with up to 94% enantiomer excess (ee). The catalyst type and the substrate scope were broadened using this methodology.

## 1. Introduction

Maleimide scaffolds are promising skeletons, which have been widely found in many natural alkaloids and bioactive pharmaceuticals [1,2,3,4,5,6,7,8]. Additionally, maleimide could be transformed into many important heterocyclic frameworks such as succinimides, pyrrolidines, and 2-pyrrolidones. Thus, intensive attention has been focused on the synthesis and modification of maleimide derivatives. In general, functionalization of maleimides mainly takes place on the double bond of maleimide through Michael addition, oxidative coupling, cycloaddition reaction, etc. [9,10,11,12,13,14,15,16,17,18,19,20,21,22,23,24]. Accordingly, enantioselective Michael addition of maleimides has been well established with maleimides as Michael acceptors [25,26,27]. Although maleimides are often used as electrophiles, their application as nucleophiles or Michael donors for the construction of maleimide-containing compounds is limited [28,29,30].

In 2021, Mori’s group first reported asymmetric Michael addition of α-aminomaleimides as Michael donors to β-nitroolefins using Cinchona alkaloid as the organocatalyst [31]. Furthermore, density functional theory (DFT) was employed in research to improve the enantioselectivity of the adduct and to reveal the mechanism of stereochemistry. Through DFT calculation, the author predicted that increasing the size of the *N* substituent of the maleimide could be favorable for stereo control. As expected, the ee value was significantly increased by 16% when *N*-Me was substituted with an *N*-*^i^*Bu group of the α-aminomaleimide in this asymmetric reaction.

To the best of our knowledge, only the above one item of literature involves asymmetric Michael addition using α-aminomaleimides as nucleophiles. Therefore, it is important to develop this reaction by using new types of catalysts. Takemoto’s catalyst is the commercially available chiral organocatalyst, which was first synthesized by Takemoto in 2003 [32]. Subsequently, they were widely used in various diastereoselective and enantioselective reactions [33,34,35,36,37,38,39,40]. In this paper, we offer the first reports on the enantioselective Michael addition of α-aminomaleimides and nitroolefins by employing Takemoto-type catalysts **1a**–**1h** bearing (thio)urea or squareamide moiety (Figure 1).

## 2. Results and Discussion

We first applied the Takemoto-type catalysts **1a**–**1h** in the Michael reaction of α-aminomaleimide (**2a**) and β-nitrostyrene (**3a**) to screen the optimal catalyst. According to the optimized condition reported [31], the reaction was carried out with Et_2_O as a solvent in the presence of 10 mol% of catalysts at r.t. (Table 1). All catalysts proceeded the reaction smoothly to give the desired product **4a** in 69–78% yields with moderate to high enantioselectivities. Among of them, (*R*,*R*)-thiourea catalyst **1c** was optimal in terms of the yield and enantioselectivity (entry 3). When the thiourea moiety in catalysts **1c** was substituted with squareamide moiety in catalyst **1d**, the similar stereoselectivity was afforded (entry 4 vs. entry 3). Additionally, *N,N*-dimethyl tertiary amine of **1c** was changed to the steric bulk moieties in catalysts **1e**–**1h**, led to reduction both in the yields and ees (entries 5–8 vs. entry 3). Therefore, *N,N*-dimethyl tertiary amine proved to be crucial. Although the optical rotation values of the products were not reported in the literature, it indicated that quinine catalyzed the reaction to obtain S configuration of the major product [31]. Therefore, we repeated a quinine-catalyzed Michael reaction of α-aminomaleimide **2a** and β-nitrostyrene **3a** to obtain the *S*-adduct **4a** (entry 9). Then, the optical rotation data of two products from quinine- or **1c**-induced reaction were determined with MeOH as solvent to afford two negative values (entries 3 and 9). Accordingly, the configuration of the major product catalyzed by **1c** was proved to be S.

To further improve the enantioselectivity of the transformation, a screened catalyst **1c** was applied to the Michael reaction of α-aminomaleimide **2a** and β-nitrostyrene **3a** under the different conditions (Table 2).

First, a variety of solvents were investigated. All the reactions proceeded smoothly in the screened solvent. It is noteworthy that aprotic solvents were suitable for the reaction to give 85–93% ees (entries 1–8), while protic MeOH was unfavorable for asymmetric induction (entry 9). Among them, toluene was optimal in terms of the yield and enantioselectivity (entry 7). When the reaction temperature was lowered from r.t. to 0 °C, an improved ee of 93% was afforded (entry 10 vs. entry 7). With further temperature decreases to −10 °C and −20 °C, both enantioselectivities were increased by 1%, but showed significantly lower yields (entries 11, 12 vs. entry 10). Therefore, 0 °C was regarded to be the most suitable reaction temperature. When reducing the catalyst loading to 5 mol%, the enantioselectivity was maintained at an excellent level, but with a relatively low yield (entry 13 vs. entry 10), and 20 mol% loading offered no improvement in the asymmetric induction, albeit with a slightly improved yield (entry 14 vs. entry 10). Furthermore, diluting the reaction concentration by half was detrimental for yield and enantiocontrol (entry 15 vs. entry 10). Adding 4 Å molecular sieves (MS) led to a slightly higher ee value of 94% and increased yield (entry 16 vs. entry 10). Based on these experiments, the optimized conditions were determined to be toluene as the solvent with a 10 mol% loading of catalyst **1c** in the presence of 4Å MS (200 mg) at 0 °C.

With the optimized conditions in hand, we explored the scope and general applicability of the protocol. A wide range of substituted α-aminomaleimides and β-nitroolefins were evaluated as shown in Table 3.

All of the substrates reacted smoothly to give the corresponding products in high yields (76–86%) with good ees (81–94%). Therefore, the stereoselectivities were barely affected by the type and position of the substituents on α-aminomaleimide. However, the substituent and position on the β-nitrostyrene was found to have influence on the enantioselectivity. It can be seen that the 2-Br substituent on phenyl of nitrostyrene led to slightly decreased ee values (entries 4, 18). Compared with the data of the reported literature listed in parentheses (Table 3) [31], our screened catalyst system showed similar enantioselectivities in most reactions. Exceptionally, in the reaction with *N*-Bn substituted maleimide as Michael donor, a markedly increased ee value was obtained (entry 17), while in the reaction with β-nitro-1-naphthalene ethylene as Michael acceptor, 10% reduction in enantioselectivity was observed (entry 12) in this paper. To extend the type of Michael acceptor in the reaction of α-aminomaleimide as donor, we tried unsaturated carbonyl compounds such as methyl cinnamate **5** and chalcone **6** to substitute the β-nitroolefin. Surprisingly, neither reaction occurred under the screened condition (Figure 1).

Based on the obtained absolute configuration described above and the previously reported enantioselective Michael addition of α-aminomaleimide **2a** and β-nitrostyrene **3a** [31], a proposed transition-state model is depicted in Figure 2. β-nitrostyrene **3a** is oriented and activated by the thiourea moiety through hydrogen bonding and the NH group in **2a** is deprotoned and oriented by the tertiary amine of catalyst **1c** through another hydrogen bonding. Then, the reaction proceeds with a *Se*-face addition of α-aminomaleimide to β-nitrostyrene, affording the desired product *S*-**4a**.

## 3. Experimental

### 3.1. Chemistry

The ^1^H NMR spectra were recorded on a 500 MHz for ^1^ H and at 125 MHz for ^13^ C NMR, using CDCl_3_ as a solvent. The chemical shifts were reported in ppm, and the residual nondeuterated solvent (CHCl_3_) as internal standard (7.26 and 77.0 ppm, respectively). The splitting patterns of the signals were reported as s, singlet; d, doublet; t, triplet; q, quartet; dd, doublet of doublets; m, multiplet. High-resolution mass spectra (HRMS) were measured on a triple TOF 5600+ mass spectrometer equipped with an electrospray ionization (ESI) source in the negative-ion mode. The enantiomeric excess (ee) values of the products were determined through chiral HPLC, using Daicel Chiralpak IA columns (4.6 mm × 250 mm). The optical rotation values were determined using an automatic polarimeter. The reactions were monitored by thin layer chromatography (TLC). Purifications by column chromatography were conducted over silica gel (200–300 mesh). The organocatalysts **1a**–**1h** were purchased from Daicel Chiral Technologies (China) Co.

### 3.2. General Procedure for the Enantioselective Michael Reaction of α-Aminomaleimides and β-Nitrostyrenes

To a mixture of nitrostyrenes (0.1 mmol), maleimides (0.1 mmol) and organocatalyst 1c (0.01 mmol), toluene (1.0 mL) was added. The resulting mixture was stirred at 0 °C for 24 h (TLC). After the reaction was finished, it was directly poured into a column chromatography on silica gel with hexane/EtOAc (5:1) as eluent to afford the products **4a**–**t**. Among them, **4b**, **4c**, **4f**–**h**, **4o**, **4p**, and **4r**–**t** were the new compounds. Experimental data can be found in the Appendix A.

#### 3.2.1. (*S*)-1-Isobutyl-3-(2-nitro-1-phenylethyl)-4-(phenylamino)-1H-pyrrole-2,5-dione (**4a**)

^1^H NMR (500 MHz, CDCl_3_) δ 7.44–7.39 (m, 2H), 7.38–7.34 (m, 1H), 7.21–7.18 (m, 3H), 7.15 (d, *J* = 7.5 Hz, 2H), 6.97 (s, 1H), 6.91–6.80 (m, 2H), 5.47 (dd, *J* = 12.5, 10.0 Hz, 1H), 4.60 (dd, *J* = 12.5, 6.0 Hz, 1H), 4.29 (dd, *J* = 10.0, 6.0 Hz, 1H), 3.34 (d, *J* = 7.5 Hz, 2H), 2.04 (hept, *J* = 7.0 Hz, 1H), 0.91 (d, *J* = 7.0 Hz, 6H); [α]_D_^25^ = −6.70 (c 0.52, MeOH) (94% ee); HPLC (Chiralpak IA, hexane:*^i^*PrOH = 90:10, 1.0 mL/min, 254 nm), t_R_ = 10.2 min (minor), 20.0 min (major).

#### 3.2.2. (*S*)-3-(1-(2-Fluorophenyl)-2-nitroethyl)-1-isobutyl-4-(phenylamino)-1H-pyrrole-2,5- dione (**4b**)

^1^H NMR (500 MHz, CDCl_3_) δ 7.35–7.27 (m, 3H), 7.25–7.17 (m, 2H), 7.09–6.98 (m, 4H), 6.94–6.88 (m, 1H), 5.39 (dd, *J* = 14.0, 11.5 Hz, 1H), 4.65–4.52 (m, 2H), 3.35 (d, *J* = 7.5 Hz, 2H), 2.05 (hept, 7.0 Hz, 1H), 0.93 (d, *J* = 6.5 Hz, 6H); ^13^C NMR (125 MHz, CDCl_3_) δ 173.2, 167.5, 141.1, 136.6, 129.5, 127.2, 124.5, 124.3 (d, *J* = 14.1 Hz), 115.5, 115.3, 95.7, 75.3, 45.4, 32.9, 27.9, 20.1. HRMS (ESI) m/z: [M + Na]^+^ calcd for C_22_H_22_FN_3_O_4_Na 434.1492; found 434.1497; [α]_D_^25^ = 6.15 (c 0.72, MeOH)(92% ee); HPLC (Chiralpak IA, hexane:*^i^*PrOH = 97:3, 0.5 mL/min, 254 nm), t_R_ = 48.6 min (minor), 51.4 min (major).

#### 3.2.3. (*S*)-3-(1-(2-Chlorophenyl)-2-nitroethyl)-1-isobutyl-4-(phenylamino)-1H-pyrrole-2,5- dione (**4c**)

^1^H NMR (500 MHz, CDCl_3_) δ 7.35 (dd, *J* = 7.0, 2.0 Hz, 1H), 7.27–7.25 (m, 3H), 7.25–7.18 (m, 3H), 7.02 (s, 1H), 6.99–6.91 (m, 2H), 5.39 (dd, *J* = 13.0, 10.5 Hz, 1H), 4.69 (dd, *J* = 10.5, 4.5 Hz, 1H), 4.57 (dd, *J* = 13.0, 4.5 Hz, 1H), 3.38 (d, *J* = 7.5 Hz, 2H), 2.07 (hept, *J* = 7.0, 1H), 0.94 (dd, *J* = 6.5, 3.5 Hz, 6H); ^13^C NMR (125 MHz, CDCl_3_) δ 173.5, 167.4, 141.4, 136.6, 134.3, 133.6, 129.9, 129.7, 129.6, 129.1, 127.4, 127.0, 124.1, 95.6, 74.5, 45.4, 36.8, 27.9, 20.0; HRMS (ESI) m/z: [M + Na]^+^ calcd for C_22_H_22_ClN_3_O_4_Na 450.1197; found 450.1194; [α]_D_^25^ = 7.41 (c 0.58, MeOH) (87% ee); HPLC (Chiralpak IA, hexane:*^i^*PrOH = 90:10, 1.0 mL/min, 254 nm), t_R_ = 9.8 min (minor), 11.3 min (major).

#### 3.2.4. (*S*)-3-(1-(2-Bromophenyl)-2-nitroethyl)-1-isobutyl-4-(phenylamino)-1H-pyrrole-2,5-dione(**4d**)

^1^H NMR (500 MHz, CDCl_3_) δ 7.44 (dd, *J* = 8.0, 1.0 Hz, 1H), 7.38 (dd, *J* = 8.0, 1.5 Hz, 1H), 7.28 (dd, *J* = 7.5, 1.0 Hz, 1H), 7.25 (d, *J* = 3.5 Hz, 3H), 7.12 (td, *J* = 7.5, 1.6 Hz, 1H), 7.00 (s, 1H), 6.97–6.91 (m, 2H), 5.40 (dd, *J* = 13.0, 10.5 Hz, 1H), 4.65 (dd, *J* = 10.5, 4.5 Hz, 1H), 4.59 (dd, *J* = 13.0, 4.5 Hz, 1H), 3.39 (d, *J* = 7.5 Hz, 2H), 2.08 (hept, *J* = 13.9, 7.0 Hz, 1H), 0.95 (dd, *J* = 6.5, 4.5 Hz, 6H); [α]_D_^25^ = 5.36 (c 0.60, MeOH) (86% ee); HPLC (Chiralpak IA, hexane:^i^PrOH = 95:5, 0.8 mL/min, 254 nm), t_R_ = 19.1 min (minor), 24.0 min (major).

#### 3.2.5. (*S*)-3-(1-(3-Bromophenyl)-2-nitroethyl)-1-isobutyl-4-(phenylamino)-1H-pyrrole-2,5- dione (**4e**)

^1^H NMR (500 MHz, CDCl_3_) δ 7.50–7.38 (m, 3H), 7.33 (ddd, *J* = 8.0, 2.0, 1.0 Hz, 1H), 7.17 (d, *J* = 7.5 Hz, 2H), 7.08 (t, *J* = 8.0 Hz, 1H), 7.00 (s, 1H), 6.89 (d, *J* = 8.0 Hz, 1H), 6.82 (t, *J* = 2.0 Hz, 1H), 5.42 (dd, *J* = 13.0, 10.0 Hz, 1H), 4.59 (dd, *J* = 13.0, 6.0 Hz, 1H), 4.22 (dd, *J* = 10.0, 6.0 Hz, 1H), 3.34 (d, *J* = 7.5 Hz, 2H), 2.03 (hept, *J* = 7.0 Hz, 1H), 0.92 (d, *J* = 6.5 Hz, 6H); [α]_D_^25^ = −24.42 (c 0.65, MeOH) (92% ee); HPLC (Chiralpak AS, hexane:*^i^*PrOH = 90:10, 1.0 mL/min, 254 nm), t_R_ = 12.6 min (minor), 15.9 min (major).

#### 3.2.6. (*S*)-3-(1-(3-Fluorophenyl)-2-nitroethyl)-1-isobutyl-4-(phenylamino)-1H-pyrrole-2,5- dione (**4f**)

^1^H NMR (500 MHz, CDCl_3_) δ 7.47–7.41 (m, 2H), 7.41–7.37 (m, 1H), 7.21–7.13 (m, 3H), 7.00 (s, 1H), 6.89 (tdd, *J* = 8.5, 2.5, 1.0 Hz, 1H), 6.64 (d, *J* = 8.0 Hz, 1H), 6.60–6.51 (m, 1H), 5.41 (dd, *J* = 13.0, 10.0 Hz, 1H), 4.62 (dd, *J* = 13.0, 6.0 Hz, 1H), 4.26 (dd, *J* = 10.0, 6.0 Hz, 1H), 3.34 (d, *J* = 7.5 Hz, 2H), 2.03 (hept, *J* = 7.0 Hz, 1H), 0.92 (d, *J* = 6.5 Hz, 6H); ^13^C NMR (125 MHz, CDCl_3_) δ 173.1, 167.5, 141.0, 140.1 (d, *J* = 6.9 Hz), 136.6, 130.3 (d, *J* = 8.1 Hz), 129.6, 127.6, 125.6, 123.5, 115.2, 115.0, 114.6 (d, *J* = 21.0 Hz), 96.6, 76.8, 45.4, 39.2, 27.9, 20.0. HRMS (ESI) m/z: [M + Na]^+^ calcd for C_22_H_22_FN_3_O_4_Na 434.1492; found 434.1497; [α]_D_^25^ = −28.98 (c 0.75, MeOH) (93% ee); HPLC (Chiralpak IA, hexane:*^i^*PrOH = 90:10, 1.0 mL/min, 254 nm), t_R_ = 10.3 min (minor), 12.4 min (major).

#### 3.2.7. (*S*)-3-(1-(4-Fluorophenyl)-2-nitroethyl)-1-isobutyl-4-(phenylamino)-1H-pyrrole-2,5- dione (**4g**)

^1^H NMR (500 MHz, CDCl_3_) δ 7.45–7.41 (m, 2H), 7.40–7.36 (m, 1H), 7.16 (d, *J* = 7.5 Hz, 2H), 6.96 (s, 1H), 6.88 (dd, *J* = 12.0, 5.5 Hz, 2H), 6.86–6.80 (m, 2H), 5.39 (dd, *J* = 12.5, 10.0 Hz, 1H), 4.60 (dd, *J* = 12.5, 6.0 Hz, 1H), 4.26 (dd, *J* = 10.0, 6.0 Hz, 1H), 3.34 (d, *J* = 7.5 Hz, 2H), 2.03 (hept, *J* = 7.0 Hz, 1H), 0.92 (d, *J* = 6.5 Hz, 6H); ^13^C NMR (125 MHz, CDCl_3_) δ 167.6, 140.7, 136.7, 133.5, 129.7(d, *J* = 8.1 Hz), 129.6, 127.5, 125.5, 115.8, 115.6, 97.2, 77.1, 45.4, 38.8, 27.9, 20.0; HRMS (ESI) m/z: [M + Na]^+^ calcd for C_22_H_22_FN_3_O_4_Na 434.1492; found 434.1499; [α]_D_^25^ = −16.12 (c 0.55, MeOH) (93% ee); HPLC (Chiralpak IA, hexane:*^i^*PrOH = 90:10, 1.0 mL/min, 254 nm), t_R_ = 10.3 min (minor), 17.2 min (major).

#### 3.2.8. (*S*)-3-(1-(4-Chlorophenyl)-2-nitroethyl)-1-isobutyl-4-(phenylamino)-1H-pyrrole-2,5- dione (**4h**)

^1^H NMR (500 MHz, CDCl_3_) δ 7.45–7.41 (m, 2H), 7.40–7.36 (m, 1H), 7.19–7.13 (m, 4H), 7.00 (s, 1H), 6.82–6.75 (m, 2H), 5.38 (dd, *J* = 12.5, 10.0 Hz, 1H), 4.60 (dd, *J* = 12.5, 6.0 Hz, 1H), 4.25 (dd, *J* = 10.0, 6.0 Hz, 1H), 3.33 (d, *J* = 7.5 Hz, 2H), 2.03 (hept, *J* = 7.0 Hz, 1H), 0.91 (d, *J* = 6.5 Hz, 6H); ^13^C NMR (125 MHz, CDCl_3_) δ 173.2, 167.5, 140.8, 136.6, 136.2, 133.5, 129.6, 129.4, 129.0, 127.6, 125.6, 96.8, 76.9, 45.4, 38.9, 31.6, 27.9, 22.6, 20.04 (d, *J* = 2.9 Hz), 14.1. HRMS (ESI) m/z: [M + Na]^+^ calcd for C_22_H_22_ClN_3_O_4_Na 450.1197; found 450.1192; [α]_D_^25^ = −58.69 (c 0.67, MeOH) (93% ee); HPLC (Chiralpak IA, hexane:*^i^*PrOH = 95:5, 0.8 mL/min, 254 nm), t_R_ = 21.6 min (minor), 32.2min (major).

#### 3.2.9. (*S*)-3-(1-(4-Bromophenyl)-2-nitroethyl)-1-isobutyl-4-(phenylamino)-1H-pyrrole-2,5- dione (**4i**)

^1^H NMR (500 MHz, CDCl_3_) δ 7.46–7.36 (m, 3H), 7.35–7.29 (m, 2H), 7.16 (d, *J* = 7.5 Hz, 2H), 6.99 (s, 1H), 6.76–6.70 (m, 2H), 5.38 (dd, *J* = 13.0, 9.5 Hz, 1H), 4.60 (dd, *J* = 13.0, 6.0 Hz, 1H), 4.23 (dd, *J* = 9.5, 6.0 Hz, 1H), 3.33 (d, *J* = 7.5 Hz, 2H), 2.02 (hept, *J* = 7.0 Hz,1H), 0.91 (d, *J* = 6.5 Hz, 6H); [α]_D_^25^ = −12.33 (c 0.49, MeOH) (90% ee); HPLC (Chiralpak IA, hexane:*^i^*PrOH = 90:10, 1.0 mL/min, 254 nm), t_R_ = 11.4 min (minor), 16.6min (major)

#### 3.2.10. (*S*)-1-Isobutyl-3-(2-nitro-1-(p-tolyl)ethyl)-4-(phenylamino)-1H-pyrrole-2,5-dione (**4j**)

^1^H NMR (500 MHz, CDCl_3_) δ 7.45–7.39 (m, 2H), 7.38–7.33 (m, 1H), 7.16 (d, *J* = 7.5 Hz, 2H), 7.01 (d, *J* = 8.0 Hz, 2H), 6.92 (s, 1H), 6.76 (d, *J* = 8.0 Hz, 2H), 5.44 (dd, *J* = 12.5, 10.0 Hz, 1H), 4.58 (dd, *J* = 12.5, 6.0 Hz, 1H), 4.25 (dd, *J* = 10.0, 6.0 Hz, 1H), 3.33 (d, *J* = 7.5 Hz, 2H), 2.28 (s, 3H), 2.04 (hept, *J* = 7.0 Hz, 1H), 0.92 (d, *J* = 6.5 Hz, 6H); [α]_D_^25^ = −8.89 (c 0.51, MeOH) (93% ee); HPLC (Chiralpak IA, hexane:*^i^*PrOH = 90:10, 1.0 mL/min, 254 nm), t_R_ = 9.5min (minor), 14.9min (major).

#### 3.2.11. (*S*)-1-Isobutyl-3-(1-(4-methoxyphenyl)-2-nitroethyl)-4-(phenylamino)-1H-pyrrole-2,5-dione (**4k**)

^1^H NMR (500 MHz, CDCl_3_) δ 7.42 (dd, *J* = 10.0, 5.0 Hz, 2H), 7.35 (dd, *J* = 8.5, 6.5 Hz, 1H), 7.15 (d, *J* = 7.5 Hz, 2H), 6.93 (s, 1H), 6.83–6.77 (m, 2H), 6.76–6.70 (m, 2H), 5.41 (dd, *J* = 12.5, 10.0 Hz, 1H), 4.57 (dd, *J* = 12.5, 6.0 Hz, 1H), 4.23 (dd, *J* = 10.0, 6.0 Hz, 1H), 3.75 (s, 3H), 3.33 (d, *J* = 7.5 Hz, 2H), 2.04 (hept, *J* = 7.0 Hz, 1H), 0.92 (d, *J* = 6.5 Hz, 6H); [α]_D_^25^ = −30.50 (c 0.59, MeOH) (92% ee); HPLC (Chiralpak IA, hexane:*^i^*PrOH = 90:10, 1.0 mL/min, 254 nm), t_R_ = 13.2min (minor), 27.6min (major).

#### 3.2.12. (*S*)-1-Isobutyl-3-(1-(naphthalen-2-yl)-2-nitroethyl)-4-(phenylamino)-1H-pyrrole- 2,5-dione (**4l**)

^1^H NMR (500 MHz, CDCl_3_) δ 7.84 (d, *J* = 8.0 Hz, 1H), 7.77 (d, *J* = 8.0 Hz, 1H), 7.51 (d, *J* = 6.5 Hz, 1H), 7.45 (ddd, *J* = 8.0, 7.0, 1.0 Hz, 1H), 7.41 (t, *J* = 8.0 Hz, 1H), 7.28 (ddd, *J* = 8.5, 7.0, 1.5 Hz, 1H), 7.11 (d, *J* = 8.5 Hz, 1H), 7.06 (t, *J* = 7.5 Hz, 1H), 7.01–6.94 (m, 3H), 6.91 (d, *J* = 7.5 Hz, 2H), 5.60 (dd, *J* = 13.0, 11.0 Hz, 1H), 5.11 (dd, *J* = 11.0, 4.0 Hz, 1H), 4.56 (dd, *J* = 13.0, 4.0 Hz, 1H), 3.42 (d, *J* = 7.5 Hz, 2H), 2.11 (hept, *J* = 7.0 Hz, 2H), 0.97 (dd, *J* = 6.5, 4.5 Hz, 6H); [α]_D_^25^ = −14.13 (c 0.66, MeOH) (92% ee); HPLC (Chiralpak IA, hexane:*^i^*PrOH = 90:10, 1.0 mL/min, 254 nm), t_R_ = 12.2min (minor), 28.2min (major).

#### 3.2.13. (*S*)-1-Isobutyl-3-(2-nitro-1-(thiophen-2-yl)ethyl)-4-(phenylamino)-1H-pyrrole-2,5- dione (**4m**)

^1^H NMR (500 MHz, CDCl_3_) δ 7.44 (dd, *J* = 10.5, 5.0 Hz, 2H), 7.37 (t, *J* = 7.5 Hz, 1H), 7.23 (d, *J* = 7.5 Hz, 2H), 7.16 (dd, *J* = 5.0, 1.0 Hz, 1H), 7.00 (s, 1H), 6.87 (dd, *J* = 5.0, 3.5 Hz, 1H), 6.65 (d, *J* = 3.5 Hz, 1H), 5.33 (dd, *J* = 12.5, 10.0 Hz, 1H), 4.63 (dd, *J* = 12.5, 5.5 Hz, 1H), 4.52 (dd, *J* = 10.0, 5.5 Hz, 1H), 3.34 (d, *J* = 7.5 Hz, 2H), 2.04 (hept, *J* = 7.0 Hz,1H), 0.92 (d, *J* = 6.5 Hz, 6H); [α]_D_^25^ = −14.13 (c 0.53, MeOH) (92% ee); HPLC (Chiralpak IA, hexane:*^i^*PrOH = 90:10, 1.0 mL/min, 254 nm), t_R_ = 12.2min (minor), 28.2min (major).

#### 3.2.14. (*S*)-3-((4-Chlorophenyl)amino)-1-isobutyl-4-(2-nitro-1-phenylethyl)-1H-pyrrole- 2,5-dione (**4n**)

^1^H NMR (500 MHz, CDCl_3_) δ 7.38 (d, *J* = 8.0 Hz, 2H), 7.28 (d, *J* = 9.0 Hz, 2H), 7.10 (d, *J* = 8.0 Hz, 2H), 6.96–6.92 (m, 3H), 5.62–5.48 (m, 1H), 4.61 (dd, *J* = 13.0, 5.5 Hz, 1H), 4.30 (dd, *J* = 10.5, 5.5 Hz, 1H), 3.36 (d, *J* = 7.5 Hz, 2H), 2.06 (hept, *J* = 7.0 Hz, 1H), 0.94 (d, *J* = 6.5 Hz, 6H); [α]_D_^25^ = −19.26 (c 0.65, MeOH) (92% ee); HPLC (Chiralpak IA, hexane:*^i^*PrOH = 95:5, 0.8 mL/min, 254 nm), t_R_ = 26.7 min (minor), 49.6 min (major).

#### 3.2.15. (*S*)-3-((4-Chlorophenyl)amino)-1-isobutyl-4-(1-(2-chlorophenyl)-2-nitroethyl)-1H- pyrrole-2,5-dione (**4o**)

^1^H NMR (500 MHz, CDCl_3_) δ 7.76–7.69 (m, 1H), 7.68–7.64 (m, 1H), 7.59 (t, *J* = 6.0 Hz, 4H), 7.34 (s, 1H), 7.26 (d, *J* = 8.5 Hz, 2H), 5.86–5.69 (m, 1H), 5.02 (dd, *J* = 11.0, 4.5 Hz, 1H), 4.91 (dd, *J* = 13.0, 4.5 Hz, 1H), 3.74 (d, *J* = 7.5 Hz, 2H), 2.43 (hept, *J* = 7.0 Hz, 1H), 1.30 (dd, *J* = 6.5, 4.5 Hz, 6H); ^13^C NMR (125 MHz, CDCl_3_) δ 173.4, 167.3, 141.3, 135.2, 134.1, 133.6, 132.7, 129.8, 129.2, 129.3, 127.5, 125.5, 96.4, 74.4, 45.4, 36.8, 27.9, 20.0; HRMS (ESI) m/z: [M + Na]^+^ calcd for C_22_H_21_Cl_2_N_3_O_4_Na 484.0807; found 484.0813; [α]_D_^25^ = 3.64 (c 0.50, MeOH) (89% ee); HPLC (Chiralpak IA, hexane:*^i^*PrOH = 90:10, 1.0 mL/min, 254 nm), t_R_ = 10.9min (minor), 14.2min (major).

#### 3.2.16. (*S*)-3-((3-Chlorophenyl)amino)-1-isobutyl-4-(2-nitro-1-phenylethyl)-1H-pyrrole- 2,5-dione (**4p**)

^1^H NMR (500 MHz, CDCl_3_) δ 7.35 (d, *J* = 7.0 Hz, 2H), 7.28 (m, 3H), 7.13 (s, 1H), 7.06 (d, *J* = 6.5 Hz, 1H), 6.99–6.89 (m, 3H), 5.58–5.44 (m, 1H), 4.62 (dd, *J* = 13.0, 5.5 Hz, 1H), 4.31 (dd, *J* = 9.5, 5.5 Hz, 1H), 3.36 (d, *J* = 7.5 Hz, 2H), 2.06 (hept, *J* = 7.0 Hz, 1H), 0.94 (d, *J* = 6.5 Hz, 6H); ^13^C NMR (126 MHz, CDCl_3_) δ 173.0, 167.5, 140.1, 138.2, 137.0, 135.2, 130.5, 129.1, 128.0, 127.8, 127.3, 125.2, 123.2, 98.9, 45.4, 39.9, 27.9, 20.0; HRMS (ESI) m/z: [M + Na]^+^ calcd for C_22_H_22_ClN_3_O_4_Na 450.1197; found 450.1190; [α]_D_^25^ = −30.83 (c 0.71, MeOH) (93% ee); HPLC (Chiralpak IA, hexane:*^i^*PrOH = 90:10, 1.0 mL/min, 254 nm), t_R_ = 10.2min (minor), 16.0min (major).

#### 3.2.17. (*S*)-1-Benzyl-3-(2-nitro-1-phenylethyl)-4-(phenylamino)-1H-pyrrole-2,5-dione (**4q**)

^1^H NMR (500 MHz, CDCl_3_) δ 7.43–7.38 (m, 2H), 7.39–7.32 (m, 5H), 7.31–7.27 (m, 1H), 7.21–7.17 (m, 3H), 7.14 (d, *J* = 7.5 Hz, 2H), 6.94 (s, 1H), 6.88–6.82 (m, 2H), 5.44 (dd, *J* = 12.5, 9.5 Hz, 1H), 4.69 (s, 2H), 4.64 (dd, *J* = 12.5, 6.0 Hz, 1H), 4.28 (dd, *J* = 9.5, 6.0 Hz, 1H); [α]_D_^25^ = −16.15 (c 0.62, MeOH) (90% ee); HPLC (Chiralpak IA, hexane:*^i^*PrOH = 90:10, 1.0 mL/min, 254 nm), t_R_ = 19.6min (minor), 24.2 min (major).

#### 3.2.18. (*S*)-1-Benzyl-3-(1-(2-bromophenyl)-2-nitroethyl)-4-(phenylamino)-1H-pyrrole-2,5- dione (**4r**)

^1^H NMR (500 MHz, CDCl_3_) δ 7.43 (dd, *J* = 8.0, 1.5 Hz, 1H), 7.40–7.29 (m, 6H), 7.26–7.23 (m, 4H), 7.11 (td, *J* = 7.5, 1.5 Hz, 1H), 7.00 (s, 1H), 6.95–6.90 (m, 2H), 5.40 (dd, *J* = 13.0, 10.5 Hz, 1H), 4.74 (dd, *J* = 21.0, 13.5 Hz, 2H), 4.66 (dd, *J* = 10.5, 4.5 Hz, 1H), 4.61 (dd, *J* = 13.0, 4.5 Hz, 1H); ^13^C NMR (125 MHz, CDCl_3_) δ 172.9, 166.9, 141.8, 136.6, 136.3, 135.8, 133.2, 130.0, 129.8, 129.3, 128.8, 128.2, 127.8, 127.1, 124.3, 124.1, 96.4, 74.5, 41.7, 39.4, 29.7; HRMS (ESI) m/z: [M + Na]^+^ calcd for C_25_H_20_BrN_3_O_4_Na 528.0535; found 528.0541; [α]_D_^25^ = 36.0 (c 0.69, MeOH) (84% ee); HPLC (Chiralpak IA, hexane:*^i^*PrOH = 90:10, 1.0 mL/min, 254 nm), t_R_ = 18.9min (minor), 23.1min (major).

#### 3.2.19. (*S*)-1-Benzyl-3-(1-(3-bromophenyl)-2-nitroethyl)-4-(phenylamino)-1H-pyrrole-2,5- dione (**4s**)

^1^H NMR (500 MHz, CDCl_3_) δ 7.50–7.38 (m, 3H), 7.38–7.27 (m, 6H), 7.15 (d, *J* = 7.5 Hz, 2H), 7.07 (t, *J* = 8.0 Hz, 1H), 6.98 (s, 1H), 6.87 (d, *J* = 8.0 Hz, 1H), 6.81 (t, *J* = 2.0 Hz, 1H), 5.39 (dd, *J* = 13.0, 9.5 Hz, 1H), 4.69 (s, 2H), 4.63 (dd, *J* = 13.0, 6.0 Hz, 1H), 4.22 (dd, *J* = 9.5, 6.0 Hz, 1H); ^13^C NMR (125 MHz, CDCl_3_) δ 172.5, 166.9, 141.4, 140.0, 136.5, 136.2, 131.2, 130.8, 130.4, 129.7, 128.7, 128.2, 127.9, 127.8, 126.4, 125.9, 122.7, 97.0, 41.6, 39.2, 29.7; HRMS (ESI) m/z: [M + Na]^+^ calcd for C_25_H_20_BrN_3_O_4_Na 528.0535; found 528.0543; [α]_D_^25^ = −11.22 (c 0.63, MeOH) (90% ee); HPLC (Chiralpak IA, hexane:*^i^*PrOH = 90:10, 1.0 mL/min, 254 nm), t_R_ = 18.7min (minor), 21.6min (major).

#### 3.2.20. (*S*)-1-Benzyl-3-(1-(4-bromophenyl)-2-nitroethyl)-4-(phenylamino)-1H-pyrrole-2,5- dione (**4t**)

^1^H NMR (500 MHz, CDCl_3_) δ 7.40 (dd, *J* = 17.0, 7.0 Hz, 3H), 7.36–7.27 (m, 7H), 7.15 (d, *J* = 6.5 Hz, 2H), 7.02 (s, 1H), 5.34 (t, *J* = 11.0 Hz, 1H), 4.76–4.57 (m, 3H), 4.23 (d, *J* = 5.0 Hz, 1H); ^13^C NMR (125 MHz, CDCl_3_) δ 172.5, 167.0, 141.2, 136.7, 136.5, 136.1, 132.0, 129.7, 129.6, 128.7, 128.2, 127.8, 127.7, 125.7, 121.7, 97.1, 41.6, 39.0, 29.7; HRMS (ESI) m/z: [M + Na]^+^ calcd for C_25_H_20_BrN_3_O_4_Na 528.0535; found 528.0546; [α]_D_^25^ = −27.63 (c 0.72, MeOH) (90% ee); HPLC (Chiralpak IA, hexane:*^i^*PrOH = 90:10, 0.5 mL/min, 254 nm), t_R_ = 42.9min (minor), 45.5min (major).

## 4. Conclusions

In summary, we described the first Takemoto-type catalyst to promote the enantioselective Michael addition of α-aminomaleimides and β-nitroolefins. The α-aminomaleimides were used as nucleophiles rather than electrophiles in this transformation to create the desired maleimide-containing adducts with high enantioselectivity (up to 94% ee). Moreover, we used our optimized conditions to expand upon the substrate scope of this reaction. Further study of α-aminomaleimides as donors in Michael additions with other acceptors is under way.

## Data Availability

Not applicable.

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
