# Peer review of "Organocatalytic Enantioselective Michael Reaction of Aminomaleimides with Nitroolefins Catalyzed by Takemoto’s Catalyst"

_molecules, 2022, doi:10.3390/molecules27227787_

Round 1

Reviewer 1 Report

The review paper is written very well, it is clear and concise. I appreciate authors for screening and optimizing different solvents. I have few comments included for the betterment of the paper.

1. Can authors comment, on the ability of introducing other functional group(s), in addition to unsaturated nitro. for example unsaturated ester or unsaturated carbonyl compound.

2. I recommend to the authors, include proposed mechanistic diagram.

Author Response

point1: Can authors comment, on the ability of introducing other functional group(s), in addition to unsaturated nitro. for example unsaturated ester or unsaturated carbonyl compound.

Response 1: Thank you for your suggestion, we chosed methyl cinnamate and chalcone as Michael acceptor, which were reacted with α‑aminomaleimide 2a. We have added the result in the manuscript as follows:

 To extend the type of Michael acceptor in the reaction of α‑aminomaleimide as donor, we tried unsaturated carbonyl compounds such as methyl cinnamate 5 and chalcone 6 to substitute the β-nitroolefin. Surprisingly, neither reaction occured under the screened condition (Scheme 1).

            Scheme 1. Michael reaction of α‑aminomaleimide 2a with unsaturated carbonyl compounds

point 2: I recommend to the authors, include proposed mechanistic diagram.

Response 2: We have proposed a plausible mechanism in manuscript as follows:

Based on the obtained absolute configuration described above and previously reported enantioselective Michael addition of α‑aminomaleimide 2a and β-nitrostyrene 3a [31], a proposed transition-state model is depicted in Scheme 2. β-nitrostyrene 3a is oriented and activated by the thiourea moiety through hydrogen bonding and the NH group in 2a is deprotoned and oriented by the tertiary amine of catalyst 1c through another hydrogen bonding. Then, the reaction proceeds with a Se-face addition of α‑aminomaleimide to β-nitrostyrene, affording the desired product S-4a.

Scheme 2. Proposed stereochemical model.

Reviewer 2 Report

The entitled manuscript “Oragnocatalytic Enantioselective Michael Reaction of Ami-2 nomaleimides with Nitroolefins Catalyzed by Takemoto’s cat-3 alyst” by Jin and coworker were reviewed and it is recommend to revise before acceptance. Some points were found and should be taken into considerations and/or improvement.

The introduction is in a medium range and could be extended to further include more details about the topic, such as DFT (if possible), Takemoto catalyst, etc.

Lines 40: catalysts 1a–1h bearing (thio)urea/squareaminde-tertiary amine moiety …, please make changes to for clearance.

Figure 1, please check the structure of 1d.

Line 66-67: remove unrequired spaces and in the entire manuscript.

Table 2: Only selected aprotic solvents were illustrated, it is important to show few trials with protic solvents, e.g. MeOH,

Table 3: R3 is missing

Good Luck

Author Response

The entitled manuscript “Oragnocatalytic Enantioselective Michael Reaction of Aminomaleimides with Nitroolefins Catalyzed by Takemoto’s catalyst” by Jin and coworker were reviewed and it is recommend to revise before acceptance. Some points were found and should be taken into considerations and/or improvement.

Point 1:The introduction is in a medium range and could be extended to further include more details about the topic, such as DFT (if possible), Takemoto catalyst, etc.

Response 1:  We have added the description of improvement ee values through DFT calculation:

 Through DFT calculation, the author predicted that increasing the size of N substituent of the maleimide could be favor to stereo control. As expected, ee value was obviously increased by 16% when N-Me was substituted with an N-iBu group of the α-aminomaleimide in this asymmetric reaction.  

We have added the introduction of Takemoto catalyst and the relevant reference 32-40: 

Takemoto’s catalyst is the commercially available chiral organocatalyst, which was first synthesized by Takemoto in 2003 [32]. Then, they were widely used in various diastereoselective and enantioselective reactions [33–40].

Point 2:Lines 40: catalysts 1a–1h bearing (thio)urea/squareaminde-tertiary amine moiety…, please make changes to for clearance.

Response 2: This unclear description has been revised as: Takemoto-type catalysts 1a–1h bearing (thio)urea or squareaminde moiety.

Point 3: Figure 1, please check the structure of 1d.

Response 3: The error has been revised.

Point 4: Line 66-67: remove unrequired spaces and in the entire manuscript.

Response 4: The alignment formats in the entire manuscript have been modified

Point 5:Table 2: Only selected aprotic solvents were illustrated, it is important to show few trials with protic solvents, e.g. MeOH,

Response 5: Thank you for your suggestion. We have done the experiment with MeOH as solvent to get the desired product with 23%ee. We have added this data to Table 2 and illustrated in the manuscript as follow: It’s noteworth that aprotic solvents were suitable for the reaction to give 85-93% ees(entries 1-8), while protic MeOH was unfavorable for asymmetric induction (entry 9).  

Point 6: Table 3: R3 is missing

Response 6: R3 in Table 3 has been added
